# Men Also Do Laundry: Multi-Attribute Bias Amplification

**Dora Zhao**
Sony AI, New York
dora.zhao@sony.com

**Jerone T.A. Andrews**
Sony AI, Tokyo

**Alice Xiang**
Sony AI, New York

## Abstract

As computer vision systems become more widely deployed, there is growing concern from both the research community and the public that these systems are not only reproducing but also amplifying harmful social biases. The phenomenon of bias amplification, which is the focus of this work, refers to models amplifying inherent training set biases at test time. Existing metrics measure bias amplification with respect to single annotated attributes (e.g., `computer`). However, several visual datasets consist of images with multiple attribute annotations. We show models can exploit correlations with multiple attributes (e.g., {`computer`, `keyboard`}), which are not accounted for by current metrics. In addition, we show current metrics can give the impression that minimal or no bias amplification has occurred, as they involve aggregating over positive and negative values. Further, these metrics lack a clear desired value, making them difficult to interpret. To address these shortcomings, we propose a new metric: Multi-Attribute Bias Amplification. We validate our metric through an analysis of gender bias amplification on the COCO and imSitu datasets. Finally, we benchmark bias mitigation methods using our proposed metric, suggesting possible avenues for future bias mitigation efforts.

## 1 Introduction

Despite their intent to faithfully depict the world, visual datasets are undeniably subject to historical and representational biases [28, 13]. Left unchecked, dataset biases are invariably learned by models, especially when they are sources of efficient features for supervised learning on a given dataset [5]. For example, an image captioning model can learn to generate gendered captions by exploiting contextual cues without ever "looking" at the person in the image [10]. Reliance on spurious correlations is undesirable since these learned associations do not always hold [24, 6]. More significantly, these associations risk not only perpetuating harmful social biases but also *amplifying* them [36].

The phenomenon of *bias amplification* refers to when a model compounds the inherent biases of its training set at test time [36]. Bias amplification has been studied across many tasks [36, 21, 31, 4, 12, 18, 29, 11, 30, 22]. Following the work of Zhao *et al.* [36], we focus on multi-label classification, using the Common Objects in Context (COCO) [19] and imSitu [34] datasets as examples. While there are metrics [36, 29, 30] that measure bias amplification in multi-label classification, they only consider the amplification that occurs between a single annotated attribute (e.g., `computer`) and a group (e.g., `female`). However, existing large-scale visual datasets often have multiple annotated attributes per image (e.g., {`computer`, `keyboard`}). Models can thus leverage correlations between a group and either single or multiple attributes simultaneously.

**Multi-attribute bias amplification metric.** In our work, we propose Multi-Attribute Bias Amplification that extends Zhao *et al.* [36] and Wang and Russakovsky [29]'s previously proposed metrics. Our new metric evaluates bias amplification arising from single and multiple attributes. We are the first to study multi-attribute bias amplification, highlighting that models exploit cor-

2022 Trustworthy and Socially Responsible Machine Learning (TSRML 2022) co-located with NeurIPS 2022.

relations between multiple attributes and group labels. We also address the issue that aggregated bias amplification metrics include summing positive and negative values. These values can cancel each other out, ostensibly presenting a smaller amount of amplification than what exists. Finally, as opposed to prior metrics that lack a clear ideal value, our metric is more interpretable.

**Gender bias amplification analysis.** Using our metric, we compare the performance of multi-label classifiers trained on COCO and im-Situ. Consistent with prior work [36, 30], we use gender expression bias amplification as a case study. As shown in Fig. 1, in imSitu, individually the verb `unloading` and location `indoors` are skewed `male`. However, when considering {unloading, indoors} in conjunction, the dataset is actually skewed `female`. Significantly, men tend to be pictured unloading *packages* outdoors whereas women are pictured unloading *laundry* or *dishes* indoors. Even when men are pictured indoors, they are pictured unloading *boxes* or *equipment* as opposed to un-

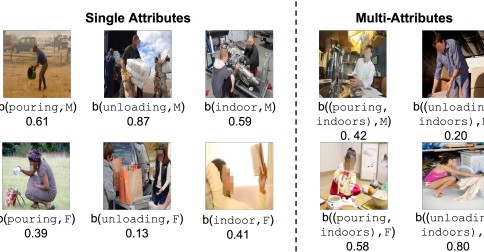

Figure 1: We provide bias scores (i.e., gender ratio) for verbs `pouring` and `unloading` as well as location `indoors` in imSitu. While the dataset is skewed male for the single attributes, the multi-attributes (e.g., {pouring, indoors}) are skewed female (F).

loading *laundry* or *dishes*. From our experiments, we find on average that bias amplification from single attributes is smaller than that from multi-attributes. Thus, if we were only to consider individual attributes, not only would we obscure the nuance of understanding multi-attributes provide but also potentially understate bias amplification.

**Benchmarking bias mitigation methods.** Finally, we benchmark mitigation methods [36, 31, 2]. We are the first to show mitigation methods for single attribute bias can actually increase multi-attribute bias amplification. This emphasizes the importance of our metric as the magnitude of amplification is likely underreported using existing metrics.

## 1.1  Related work

**Measuring bias amplification.** Dataset biases can lead to downstream harms when models are trained on these datasets. Machine learning models can not only reproduce but also amplify them [36]. Zhao *et al.* [36] propose a metric for bias amplification that measures the difference in object co-occurrences from the training to predicted distribution. Building on this work, Wang and Russakovsky [29] proposed "directional bias amplifcation", which disentangles bias arising from the attribute versus group prediction. An alternative line of work [30, 11] uses *leakage*—the change in a classifier's ability to predict group membership from the training data to predictions. Our work extends the co-occurrence-focused metrics [36, 29]. We are the first to consider how multi-attributes can impact bias amplification and propose a novel metric for quantifying this phenomenon.

**Mitigating bias amplification.** There are several proposed methods for mitigating bias amplification at both the dataset and model level. At the dataset level, a popular method [21, 25, 27] has been to use generative adversarial networks (GANs) to create synthetic examples for augmenting the training set. In the NLP domain, prior works [14, 32] also use generated counterfactuals to reduce biases. Alternatively, recent work [1] proposed a more sophisticated re-sampling strategies to address spurious correlations with objects, which models often leverage to amplify bias. At the model level, existing mitigation strategies include corpus-level constraints [36], adversarial debiasing [30], and domain independent training [31]. We refer the reader to Wang *et al.* [31] for a comprehensive survey of different mitigation methods. In our work, we do not aim to propose a new mitigation strategy; rather, we show existing methods do not mitigate bias amplification from multiple attributes.

## 2  Multi-attribute bias amplification

We outline our notation and introduce our new metric *multi-attribute bias amplification*. Throughout this paper, we scale all metrics by a factor of 100.

**Notation.** We denote by $\mathcal{G} = \{g_1, \ldots, g_t\}$ and $\mathcal{A} = \{a_1, \ldots, a_n\}$ a set of $t$ group membership labels and a set of $n$ attributes, resp., where $g_i \in \mathcal{G}$ denotes group membership and $a_i \in \mathcal{A}$ denotes the absence (i.e., $a_i = 0$) or presence (i.e., $a_i = 1$) of attribute $i$. Further, let $x \in \mathbb{R}^d$ and $y = [g_1, \ldots, g_t, a_1, \ldots, a_n] \in \{0, 1\}^{t+n}$ denote an image and its ground truth labels, resp., sampled from dataset $\mathcal{D}$.

We define $\mathcal{M} = \{m_1, \ldots, m_\ell\}$ as a set of $\ell$ sets, containing all possible combinations of attributes, where $m_i$ is a set of attributes and $\in \{1, \ldots, n\}$. Note $m \in \mathcal{M}$ if and only if $C(m, g) \geq 1$ in both the ground-truth training set and test set, where $C(m, g)$ is the number of times $m$ and $g$ co-occur.

We extend Zhao *et al.* [36]'s *bias score* to a multi-attribute setting such that the bias score of a set of attributes $m \in \mathcal{M}$ w.r.t. group $g \in \mathcal{G}$ is defined as $b(m, g) = \frac{C(m,g)}{\sum_{g'} C(m,g')}$.

**Definition 1: Undirected multi-attribute bias amplification.** Let $\mathcal{G}^+ = \{g \in \mathcal{G} \mid b(m, g) > |\mathcal{G}|^{-1}\}$ denote a set of group membership labels that are positively correlated with a set of attributes. We define our *undirected* multi-attribute bias amplification metric as:

$$\text{Multi}_{\text{MALS}} = X, \text{Var}(\Delta_{gm}) \tag{1}$$

where $X = \frac{1}{|\mathcal{M}'|} \sum_{g \in \mathcal{G}^+} \sum_{m \in \mathcal{M}'} |\Delta_{gm}|$ and $\Delta_{gm} = \widehat{b}(m_i, g_i) - b(m_i, g_i)$. Here $\widehat{b}(m, g)$ is the bias score from the attribute and group label test set predictions, whereas $b(m, g)$ is the bias score from the attribute and group label training set ground truths. $\text{Multi}_{\text{MALS}}$ measures both the mean and variance over the change in bias score from the training set ground truths to test set predictions.

**Definition 2: Directional multi-attribute bias amplification.** Let $\widehat{m}$ and $\widehat{g}$ denote a model's prediction for attribute group, $m$, and group membership, $g$, respectively. We define our *directional* multi-attribute bias amplification metric that takes into account the direction of bias as:

$$\text{Multi}_{\rightarrow} = X, \text{Var}(\Delta_{gm}) \tag{2}$$

where

$$X = \frac{1}{|\mathcal{G}||\mathcal{M}|} \sum_{g \in \mathcal{G}} \sum_{m \in \mathcal{M}} y_{gm} |\Delta_{gm}| + (1 - y_{gm}) |-\Delta_{gm}|,$$

$$y_{gm} = \mathbb{1} \left[ P(g = 1, m = 1) > P(g = 1)P(m = 1) \right], \text{ and}$$

$$\Delta_{gm} = \begin{cases} P(\widehat{m} = 1 | g = 1) - P(m = 1 | g = 1) & \text{(if measuring } G \rightarrow M) \\ P(\widehat{g} = 1 | m = 1) - P(g = 1 | m = 1) & \text{(if measuring } M \rightarrow G). \end{cases}$$

Unlike $\text{Multi}_{\text{MALS}}$, $\text{Multi}_{\rightarrow}$ captures both positive and negative correlations, i.e., $\text{Multi}_{\rightarrow}$ iterates over all $g \in \mathcal{G}$ regardless of whether $b(m, g) > |\mathcal{G}|^{-1}$. Moreover, $\text{Multi}_{\rightarrow}$ takes into account the base rates for group membership and disentangles bias amplification arising from the group influencing the attribute(s) prediction (i.e., $\text{Multi}_{G \rightarrow M}$), as well as bias amplification from the attribute(s) influencing the group prediction (i.e., $\text{Multi}_{M \rightarrow G}$).

**Relation to existing metrics.** In Eqs. (1) and (2), if we: (i) enforce $m_i \in \mathcal{M} \iff |m_i| = 1$; (ii) do not take the absolute value of $\pm \Delta_{gm}$ when computing $X$; and (iii) only report $X$, then our metric $\text{Multi}_{\text{MALS}}$ reduces to Zhao *et al.* [36]'s undirected *single-attribute* bias amplification metric $\text{BA}_{\text{MALS}}$ and $\text{Multi}_{\rightarrow}$ reduces to Wang and Russakovsky's [29] directional *single-attribute* bias amplification metric $\text{BA}_{\rightarrow}$. See Appendix A for a formal definition of $\text{BA}_{\text{MALS}}$ and $\text{BA}_{\rightarrow}$.

## 3 Comparison with existing metrics

We compare our proposed metrics $\text{Multi}_{\text{MALS}}$ and $\text{Multi}_{\rightarrow}$ to $\text{BA}_{\text{MALS}}$ [36] and $\text{BA}_{\rightarrow}$ [29].

**Setup.** We perform multi-label classification on a manipulated MNIST. For simplicity, we convert the task to binary classification [7], such that half the digits are arbitrarily assigned to group 0 or 1. For the attributes, per image, we set a combination of three corner pixels to white. We train a LeNet-5 [17], average over 5 random group assignments, and report the 95% confidence interval. See Appendix H for more details.

**Advantage 1: Our metric accounts for co-occurrences with multiple attributes.** If a model, for example, learns that the combination of $a_1$ and $a_2$, denoted $(a_1, a_2)$, are correlated with $g$, it can exploit this correlation, potentially leading to bias amplification. By limiting the measurement to single attribute co-occurrences, $BA_{MALS}$ and $BA_\rightarrow$ do not account for amplification arising from co-occurrences with multiple attributes.

To illustrate this, we manipulate MNIST so that the dataset is perfectly balanced wrt single attributes, i.e., $b(a_i, g) = 0.5$ ($\forall i \in [1, 2, 3]$), but skewed for multiple attributes. For example, although $b(a_1, g) = b(a_2, g) = 0.5$, the bias score for the combination of $(a_1, a_2)$ and $g_1$ is $b((a_1, a_2), g_1) = 0.8$. Bias amplification is $0.0 \pm 0.0$ for all three single-attribute metrics. However, the bias scores calculated wrt multiple attributes increased (see Tbl. 1). Therefore, bias has been amplified but is not being captured by existing metrics. Significantly, by iterating over all groups in $\mathcal{M}$ our proposed metric accounts for amplifi-

| Multi-attributes | $b(a, g)$ | $\widehat{b}(a, g)$ |
|---|---|---|
| $(a_1, a_2)$ | 0.80 | $0.92 \pm 0.1$ |
| $(a_1, a_3)$ | 0.49 | $0.50 \pm 0.0$ |
| $(a_2, a_3)$ | 0.80 | $0.99 \pm 0.0$ |
| $(a_1, a_2, a_3)$ | 0.94 | $1.00 \pm 0.0$ |

Table 1: Multi-attributes bias scores of group $g = 1$ for ground-truth ($b(a, g)$) and predicted ($\widehat{b}(a, g)$) labels. We report the $95\%$ confidence interval over 5 random assignments of $g$.

cation from singular and multi-attributes. We capture attributes exhibiting amplification which may not have been previously identified. While existing metrics report amplification values close to 0, our multi-attribute metric returns $9.2 \pm 2.2$, $0.3 \pm 0.1$, $0.2 \pm 0.1$ for $Multi_{MALS}$, $Multi_{G\rightarrow M}$, and $Multi_{M\rightarrow G}$, resp.

**Advantage 2: Negative and positive values do not cancel each other out.** Existing metrics calculate bias amplification by aggregating over the difference in bias scores for each attribute. Suppose there is a dataset with two annotated attributes $a_1$ and $a_2$. It is possible $\Delta_{ga_1} \approx -\Delta_{ga_2}$ for $BA_{MALS}$ or equivalently the difference in bias scores have opposite signs for $BA_\rightarrow$. In such cases, bias amplification would be $\approx 0$, giving the impression that bias amplification is minimal to none.

More concretely, we arbitrarily set $a_1$, $a_2$, and $a_3$ in MNIST to 0 with a probability of 0.7, 0.2, and 0.4, resp. The model achieves mAP of $85.2 \pm 9.9$. One of the models results in $BA_{MALS} \approx 0.0$, suggesting no bias amplification occurred. However, the bias scores for individual attributes are $\Delta_{ga_1} = 0.61$ and $\Delta_{ga_2} = -0.60$. Prior work [29] recognized this limitation and suggested returning group-wise disaggregated pairs per attribute. But, disaggregated values are difficult to interpret and make comparing models cumbersome. Alternatively, our metric uses the absolute values of differences. Doing so ensures positive and negative bias amplifications per attribute do not cancel each other out. Further, it allows us to provide a single metric, which is more understandable than disaggregated values for all attributes.

**Advantage 3: Our metric is more interpretable.** There is a lack of intuition on what an "ideal" amplification value is. One interpretation is that smaller values are more desirable. However, lower negative values are not always ideal. First, there can exist a trade-off between performance and smaller bias amplification values. Second, high magnitude negative bias amplification may lead to erasure of certain groups. For example, in imSitu, the $b(\texttt{typing}, \texttt{F}) = 0.52$. Negative bias amplification signifies the model underpredicts ($\texttt{typing}, \texttt{F}$), which can reinforce gender stereotypes [36]. Instead, we may want to minimize the distance between the bias amplification value and 0. This interpretation offers the advantage that large negative values are also not desirable. Nonetheless, a potential dilemma occurs when interpreting two values with the same magnitude but opposite signs, which represents a value-laden decision and depends on the system's context. Differently, our proposed metric is easy to interpret. Since we use absolute differences, the ideal value is unambiguously 0. Further, reporting variance provides intuition about whether amplification is uniform across all attributes or if particular attributes are more amplified.

## 4 Evaluating multi-attribute bias amplification

We now analyze the advantages of our proposed metric. To do so, we evaluate bias amplification when group membership is balanced wrt to single attributes.

**Datasets.** In our experiments, we focus on COCO [19] and imSitu [34] as they contain multiple attributes per image and are frequently used in explorations of bias amplification [36, 30, 29]. For attributes, in COCO we consider the prediction of objects and for imSitu the prediction of the verb and location. We look at 52 objects in COCO and 361 verbs in imSitu. To balance the datasets

wrt single attributes, we iterate over each attribute $a$ and greedily oversample until the bias score $b(a, g) \in [|\mathcal{G}|^{-1} \pm \epsilon](a, g)$, where $\epsilon = 0.025$. See Appendix B for more preprocessing details. Our group membership is binary gender expression, i.e., {female, male}. Due to the lack of self-reported demographic annotations, we use third-party judgement and only consider binary labels. Relying on proxy judgements reifies the incorrect notion that gender identity can be visually inferred, and reducing gender to a binary is harmful [8, 15]. We note our metric can account for any number of groups, including intersectional groups, so long as there exists relevant annotations.

**Model.** Following prior works [30, 31, 27, 36], we use a ResNet-50 [9] pre-trained on ImageNet [23]. We replace the final layer to jointly predict the group membership and attributes. The model is trained for 50 epochs using an Adam optimizer [16]. We train five models with random seeds and report the 95% confidence interval. See Appendix C for more training details.

**Experimental analysis: Multiple attributes.** First, we examine the effect of including multiple attributes by varying the minimum size of $|m_i|$, i.e., number of attributes in a combination. In Tbl. 2, we consider when $|m_i| \geq 2$ (i.e., only combinations of multi-attributes) and $|m_i| \geq 1$ (i.e., both single and multi-attributes). Interestingly, for COCO and imSitu, Multi$_{\text{MALS}}$ is greater when $|m_i| \geq 2$ at $18.1 \pm 0.4$ and $17.6 \pm 0.4$ versus $17.3 \pm 0.4$ and $9.3 \pm 0.2$. This implies mean bias amplification from single attributes is lower than that of multiple attributes.

We also observe for imSitu, there is a large decrease in amplification in Multi$_{A \rightarrow G}$ from $|m_i| \geq 1$ to $|m_i| \geq 2$. When considering disag-

| (a) COCO | $|m_i| \geq 2$ | $|m_i| \geq 1$ |
|---|---|---|
| Multi$_{\text{MALS}}$ | $18.1 \pm 0.4, 4.4 \pm 0.1$ | $17.3 \pm 0.4, 4.1 \pm 0.1$ |
| Multi$_{M \rightarrow G}$ | $3.1 \pm 0.2, 0.1 \pm 0.0$ | $3.2 \pm 0.2, 0.2 \pm 0.0$ |
| Multi$_{G \rightarrow M}$ | $0.2 \pm 0.0, 0.0 \pm 0.0$ | $0.2 \pm 0.0, 0.0 \pm 0.0$ |

| (b) imSitu | $|m_i| \geq 2$ | $|m_i| \geq 1$ |
|---|---|---|
| Multi$_{\text{MALS}}$ | $17.6 \pm 0.4, 3.0 \pm 0.1$ | $9.3 \pm 0.2, 1.6 \pm 0.1$ |
| Multi$_{M \rightarrow G}$ | $2.2 \pm 0.2, 0.0 \pm 0.0$ | $6.9 \pm 0.1, 1.9 \pm 0.0$ |
| Multi$_{G \rightarrow M}$ | $0.1 \pm 0.0, 0.0 \pm 0.0$ | $0.1 \pm 0.0, 0.0 \pm 0.0$ |

Table 2: Multi-attribute bias amplification (mean and variance) when varying the minimum number of attributes in a combination. $|m_i| \geq 1$ includes biases from single and multi-attributes. We report 95% confidence interval over five models trained using random seeds for COCO (a) and imSitu (b).

gregated values, there is considerably higher bias from certain singular verbs, such as constructing, reading, and vacuuming, for $A \rightarrow G$ prediction. Finally, we note for single attributes, larger bias amplification occurs with attributes that co-occur with males, but more for females when considering multiple attributes.

**Experimental analysis: Raw versus absolute differences.** Now, we examine how the metrics differ when using raw differences versus absolute differences (see Tbl. 3). First, looking at BA$_{\text{MALS}}$ and Multi$_{\text{MALS}}$, we see bias amplification increases in magnitude when we add multiple attributes. Looking at the variance of Multi$_{\text{MALS}}$, we see it is considerably higher at $4.1 \pm 0.1$ compared to the variance of $0.2 \pm 0.0$ for BA$_{\text{MALS}}$. This indicates there are likely specific groups of multi-attributes where there is more bias amplification arising. Next, comparing the results for raw versus absolute differences, we observe the magnitude increases significantly when using the absolute differences for some metrics. For example, BA$_{A \rightarrow G}$ on imSitu is $0.7 \pm 0.1$ for raw differences. This value is close to zero, suggesting there is not much amplification arising from attribute to group prediction. However, BA$_{A \rightarrow G}$ is $11.9 \pm 0.1$ when using absolute difference. The large increase in magnitude suggests the

| (a) Raw | mAP | BA$_{\text{MALS}}$ | BA$_{A \rightarrow G}$ | BA$_{G \rightarrow A}$ | Multi$_{\text{MALS}}$ | Multi$_{M \rightarrow G}$ | Multi$_{G \rightarrow M}$ |
|---|---|---|---|---|---|---|---|
| COCO | $53.8 \pm 0.1$ | $-1.9 \pm 0.2$ | $-1.5 \pm 0.2$ | $-0.0 \pm 0.0$ | $-10.1 \pm 0.3$ | $-1.2 \pm 0.1$ | $0.0 \pm 0.0$ |
| imSitu | $67.0 \pm 0.1$ | $0.3 \pm 0.1$ | $0.7 \pm 0.1$ | $0.0 \pm 0.0$ | $-2.4 \pm 0.3$ | $1.0 \pm 0.1$ | $0.0 \pm 0.0$ |

| (b) Absolute | | BA$_{\text{MALS}}$ | BA$_{A \rightarrow G}$ | BA$_{G \rightarrow A}$ | Multi$_{\text{MALS}}$ | Multi$_{M \rightarrow G}$ | Multi$_{G \rightarrow M}$ |
|---|---|---|---|---|---|---|---|
| COCO | | $5.5 \pm 0.2$ | $5.0 \pm 0.2$ | $0.3 \pm 0.0$ | $17.3 \pm 0.4$ | $3.2 \pm 0.2$ | $0.2 \pm 0.0$ |
| imSitu | | $1.3 \pm 0.0$ | $11.9 \pm 0.1$ | $0.1 \pm 0.0$ | $9.3 \pm 0.2$ | $6.9 \pm 0.1$ | $0.1 \pm 0.0$ |

| (c) Variance | | BA$_{\text{MALS}}$ | BA$_{A \rightarrow G}$ | BA$_{G \rightarrow A}$ | Multi$_{\text{MALS}}$ | Multi$_{M \rightarrow G}$ | Multi$_{G \rightarrow M}$ |
|---|---|---|---|---|---|---|---|
| COCO | | $0.2 \pm 0.0$ | $0.4 \pm 0.0$ | $0.0 \pm 0.0$ | $4.1 \pm 0.1$ | $0.2 \pm 0.0$ | $0.0 \pm 0.0$ |
| imSitu | | $0.1 \pm 0.0$ | $2.5 \pm 0.0$ | $0.0 \pm 0.0$ | $1.6 \pm 0.1$ | $1.9 \pm 0.0$ | $0.0 \pm 0.0$ |

Table 3: Evaluation of three versions of the bias amplification metrics on "balanced" datasets. There are three versions of the bias amplification metrics: calculated using raw differences from training to test set (a), absolute differences (b), and variance of differences (c). We report the 95% confidence interval over 5 models trained using random seeds.

| | mAP | $\text{BA}_{\text{MALS}}$ | $\text{BA}_{A \to G}$ | $\text{BA}_{G \to A}$ | $\text{Multi}_{\text{MALS}}$ | $\text{Multi}_{M \to G}$ | $\text{Multi}_{G \to M}$ |
|---|---|---|---|---|---|---|---|
| ORIGINAL | $67.1 \pm 0.1$ | $2.5 \pm 0.1$ | $-0.3 \pm 0.1$ | $0.0 \pm 0.0$ | $\mathbf{15.1 \pm 0.1}$ | $3.3 \pm 0.1$ | $\mathbf{0.1 \pm 0.0}$ |
| OVERSAMPLING | $66.3 \pm 0.1$ | $-4.5 \pm 0.2$ | $-2.4 \pm 0.1$ | $-0.0 \pm 0.0$ | $15.7 \pm 0.4$ | $0.2 \pm 0.1$ | $0.1 \pm 0.0$ |
| RBA | $54.7 \pm 0.5$ | $\mathbf{-1.4 \pm 0.3}$ | $-6.2 \pm 0.3$ | $-0.1 \pm 0.0$ | $17.6 \pm 0.3$ | $12.4 \pm 0.9$ | $1.1 \pm 0.1$ |
| ADV | $58.1 \pm 0.1$ | $4.1 \pm 0.3$ | $0.6 \pm 0.4$ | $-0.0 \pm 0.0$ | $19.1 \pm 0.5$ | $5.4 \pm 0.8$ | $0.3 \pm 0.1$ |
| DOMIND | $\mathbf{69.6 \pm 0.1}$ | $10.2 \pm 0.9$ | $\mathbf{0.0 \pm 0.0}$ | $0.1 \pm 0.0$ | $20.3 \pm 0.6$ | $\mathbf{0.0 \pm 0.0}$ | $0.5 \pm 0.2$ |
| DATA REPAIR | $62.3 \pm 0.1$ | $-1.8 \pm 0.1$ | $-0.1 \pm 0.1$ | $\mathbf{-0.0 \pm 0.0}$ | $21.4 \pm 0.3$ | $14.0 \pm 0.1$ | $1.6 \pm 0.0$ |

Table 4: Comparison of 5 mitigation methods—OVERSAMPLING, RBA, ADV [30], DOMIND, and DATA REPAIR—against the baseline ORIGINAL on imSitu. We report the $95\%$ confidence interval over 5 models trained using random seeds. The bold values indicate the best performing method: distance to 0 for single-attribute and smallest value for multi-attribute metrics.

negative and positive values for individual attributes are collapsed to zero when taking the mean; this relays an erroneous message that there is minimal bias amplification.

**Experimental analysis: $G \to A$ bias amplification.** Finally, we find amplification from $G \to A$ prediction is consistently low. For datasets such as COCO and imSitu where the person may be small or not the image's focal point, it is possible group membership is harder to inferr and has a smaller impact on attribute prediction. We evaluate $\text{BA}_{G \to A}$ and $\text{Multi}_{G \to M}$ with images containing person bounding boxes of varying sizes. There is a Pearson's $r$ of $0.89$ and $0.92$ between bias amplification and the person bounding box size with for $\text{BA}_{G \to A}$ and $\text{Multi}_{G \to M}$, respectively (see Appendix E).

## 5 Benchmarking bias mitigation methods

Next, we consider how previously proposed bias mitigation methods [30, 36, 31, 1] perform when evaluated using multi-attribute bias amplification metrics.

**Dataset and mitigation methods.** We use imSitu as our testbed for experimentation. See Appendix F.2 for results on COCO. We derive the labels using the same process described in Sec. 4. The key difference is that we do *not* oversample to balance the attribute and group co-occurrences. Following Wang *et al.* [31], we benchmark five mitigation methods: oversampling, corpus constraints (RBA) [36], adversarial de-biasing (ADV) [30], domain independent training (DOMIND) [31], and data repair [1]. See Appendix F for more details.

**Experimental analysis: Single attribute methods do not always work for multi-attributes.** We compare the performance of mitigation methods on single (i.e., BA) versus multi-attribute metrics (i.e., Multi). All mitigation methods, save for OVERSAMPLING perform in line with or outperform ORIGINAL on single attribute metrics (see Tbl. 6). However, bias reduction for single attributes does not necessarily indicate bias has been reduced for multi-attributes. RBA reduced $\text{BA}_{\text{MALS}}$ from $2.5 \pm 0.1$ on ORIGINAL to $-1.4 \pm 0.3$, but $\text{Multi}_{\text{MALS}}$ increased from $15.1 \pm 0.1$ to $17.6 \pm 0.3$. In fact, all mitigation methods increased bias relative to ORIGINAL for $\text{Multi}_{\text{MALS}}$ and $\text{Multi}_{G \to M}$. While current mitigation methods may work for single attributes, this leads to greater amplification for multi-attributes, indicating the overall amount of bias amplification may not actually be decreasing. We provide more intuition on why this occurs in Appendix F.

**Experimental analysis: Mitigation methods struggle with $G \to A$ amplification.** Next, we consider group to attribute prediction amplification (i.e., $\text{BA}_{G \to A}$, $\text{Multi}_{G \to M}$). As shown in Tbl. 6, while the methods can mitigate $A \to G$ bias amplification, they fare worse in the opposite direction. Save for $\text{BA}_{G \to A}$ where DATA REPAIR slightly outperforms ORIGINAL, the baseline model outperforms all mitigation methods. This finding is in line with that from Wang and Russakovsky [29]. Mitigating $G \to A$ amplification, particularly for classification tasks where predicting group membership can be unnecessary and potentially harmful [29, 8, 26], is an important open direction for future exploration.

## 6 Conclusion

Our proposed metric, Multi-Attribute Bias Amplification, illustrates the need to consider multiple attributes when measuring bias amplification. For perfectly "balanced" datasets, we find bias amplification occurs w.r.t multi-attributes, regardless of whether we use raw or absolute differences. Further, we are the first to show that methods that mitigate single attribute bias can inadvertently

increase multi-attribute bias amplification. Overall, multi-attribute bias amplification provides a better understanding of the extent of bias a model introduces from training to prediction.

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

## A   Existing metrics definition

Using the notation from Sec. 2, we provide formal definitions for $\text{BiasAmp}_{\text{MALS}}$ and $\text{BiasAmp}_{\rightarrow}$. We first provide the definition for undirected bias amplification [36]:

$$\text{BiasAmp}_{\text{MALS}} = \frac{1}{|\mathcal{A}|} \sum_{\substack{g \in \mathcal{G} \\ a \in \mathcal{A}'}} \widehat{b}(a, g) - b(a, g), \tag{3}$$

where $a \in \mathcal{A}' \iff b(a, g) > |\mathcal{G}|^{-1}$. Note $\widehat{b}(a, g)$ is the bias score from the attribute and group label test set predictions, whereas $b(a, g)$ is the bias score from the attribute and group label training set ground truths.

Next, we provide the definition for directional bias amplification [29].

$$\text{BiasAmp}_{\rightarrow} = \frac{1}{|\mathcal{G}||\mathcal{A}|} \sum_{\substack{g \in \mathcal{G} \\ a \in \mathcal{A}}} y_{ga} \Delta_{ga} + (1 - y_{ga})(-\Delta_{ga}), \tag{4}$$

where

$$y_{ga} = 1\left[P(G_g = 1, A_a = 1) > P(A_a = 1)P(G_g = 1)\right]$$

$$\Delta_{ga} = \begin{cases} P(\hat{A}_a = 1 | G_a = 1) - P(A_a = 1 | G_a = 1) \\ \text{if measuring } G \rightarrow A \\ P(\hat{G}_g = 1 | A_a = 1) - P(G_g = 1 | A_a = 1) \\ \text{if measuring } A \rightarrow G \end{cases} \tag{5}$$

Here $\hat{A}$ and $\hat{G}$ denote the model's predictions for an attribute, $a$, and the group membership, $g$.

## B   Dataset preprocessing

We provide additional details on how COCO and imSitu are preprocessed.

### B.1   COCO

**Gender expression labels.** To derive the labels, for COCO, we follow Zhao *et al.* [36] and use the provided captions. We only consider objects occurring $> 100$ times with either group, leading to 52 objects in total. Following Zhao *et al.* [35], we use an expanded word set to automatically derive perceived gender expression from the captions provided in the COCO [19] dataset. When searching for gendered words, we first convert the captions to lowercase. We then use the following set of keywords to query the captions:

- Male: {"male", "boy", "man", "gentleman", "boys", "men", "males", "gentlemen", "father", "boyfriend"}
- Female: {"female", "girl", "woman", "lady", "girls", "women", "females", "ladies", "mother", "girlfriend"}

One of the five COCO captions must contain a gendered term. Further, if both a male and female gendered term appeared in the captions, the instance was discarded. This methodology matches the one used in prior work [36, 30, 1]

**Attribute labels.** We only include attributes that have occurred more than 100 times with both male and female instances. This leaves us with the following 52 attributes: {"person", "bicycle", "car", "motorcycle", "bus", "truck", "traffic light", "bench", "cat", "dog", "horse", "backpack", "umbrella",

| $|m_i|$ | $n$ | Multi$_{\text{MALS}}$ | Multi$_{M \to G}$ | Multi$_{G \to M}$ |
|---|---|---|---|---|
| 1 | 51 | $5.5 \pm 0.2$ | $5.0 \pm 0.2$ | $0.3 \pm 0.0$ |
| 2 | 428 | $16.8 \pm 0.4$ | $3.3 \pm 0.2$ | $0.3 \pm 0.0$ |
| 3 | 373 | $20.1 \pm 0.3$ | $2.8 \pm 0.2$ | $0.2 \pm 0.0$ |
| 4 | 147 | $23.2 \pm 0.5$ | $2.9 \pm 0.3$ | $0.2 \pm 0.0$ |
| 5 | 33 | $20.0 \pm 0.5$ | $2.7 \pm 0.3$ | $0.2 \pm 0.0$ |

Table 5: Multi-attribute bias amplification for varying attribute group sizes for COCO. When the group size is 3, we only consider amplification arising from a group of three attributes. Note when the group size equals 1 the metric reduces to single-attribute bias amplification.

"handbag", "tie", "suitcase", "frisbee", "skis", "sports ball", "kite", "surfboard", "tennis racket", "bottle", "wine glass", "cup", "fork", "knife", "spoon", "bowl", "banana", "sandwich", "pizza", "donut", "cake", "chair", "couch", "potted plant", "bed", "dining table", "tv", "laptop", "remote", "cell phone", "microwave", "oven", "sink", "refrigerator", "book", "clock", "vase", "teddy bear", "toothbrush"}.

### B.2 imSitu

**Gender expression labels.** For imSitu, we derive group labels using gendered terms for the agent and verbs that have occurred $> 5$ with either group. We consider the 361 verbs that have a location component, where location is a binary prediction between indoor or outdoor. To derive the gender expression labels, we use the agent annotations associated with each instance. We use the same set of keywords that we used for COCO to query male and female instances.

**Action and location labels.** From the 504 annotated verbs in imSitu, we consider only those that co-occur with a gendered agent. Further, we only include actions that have occurred more than 5 times with both male and female instances. This leaves us with 361 verbs in total.

We turn location into a binary prediction between indoor and outdoor. To derive labels, we consider the top 50 place annotations in imSitu. We manually annotate each location as either "indoor" or "outdoor," discarding locations that we were unsure about. In total, we end up with 25 location annotations that are divided into indoor and outdoor as follows:

- Outdoors: {"outdoors", "outside", "field", "street", "road", "sidewalk", "beach", "farm", "forest", "yard"}

- Indoors: {"room", "inside", "kitchen", "office", "gymnasium", "shop", "house", "hospital", "classroom", "workshop", "stage", "bed", "classroom", "living room", "bathroom"}

## C Training details

All models in this work were developed using PyTorch. The models are trained and evaluated on 1 NVIDIA T4 Tensor Core GPU with 64 GB of GPU memory and 2.5 GHz Cascade Lake 24C processors. The operating system is Linux 64-bit Ubuntu 18.04.

## D Effect of varying group size

We report the results for varying values the minimum number of attributes, $|m_i|$ in a combination. When $|m_i| \geq 1$, this includes both single and multi attributes; when $|m_i| \geq 2$, this includes only multi-attributes. In Fig. 2, in addition to these two values, we also report $|m_i| \geq 3$, $|m_i| \geq 4$, $|m_i| \geq 5$. While we do not set an upper bound for $|m_i|$ in our metric, in practice it is unlikely to find larger groups of attributes in most real-world datasets. For example, in COCO, the largest multi-attribute group we consider contains 6 attributes. Overall, we find that as values of $|m_i|$ increase, Muli$_{\text{MALS}}$ increases; however, the values for directional multi-attribute amplification stay relatively consistent. This trend is mirrored in Tbl. 5 where we focus on when $|m_i|$ is set to a constant value (e.g., $|m_i| = 2$ means we only consider multi-attribute groups of size 2).

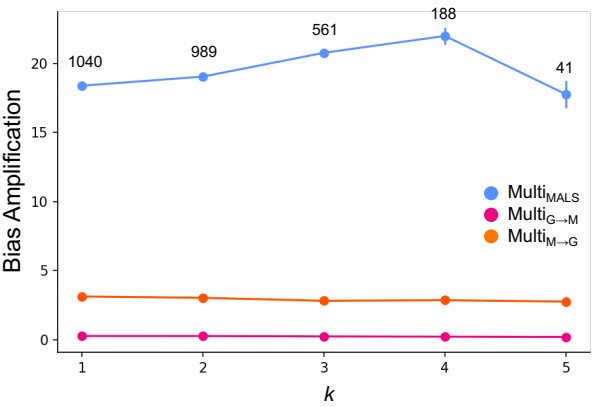

Figure 2: Multi-attribute bias amplification for different values of $|m_i|$ where at 1 we capture bias amplification for $|m_i| \geq 1$, $|m_i|$ is $k \geq 2$, and so on for COCO. Above Multi$_{\text{MALS}}$, we report the number of attribute groups for each $|m_i|$ value. We plot the average over five runs with random seeds of the model. Error bars represent the standard deviation over runs.

# E    Additional balanced dataset results

Finally, we observe amplification arising from gender expression to attribute prediction (BiasAmp$_{G \rightarrow A}$, Multi$_{G \rightarrow M}$) is consistently low. Although spurious correlations with gender exist in many parts of the image (e.g., color, contextual objects) [20], the person is the main source of gender cues. For datasets such as COCO and imSitu where the person may be small or not the image's focal point, it is possible group membership is less easily inferred and thus has a smaller impact on attribute prediction. We evaluate BiasAmp$_{G \rightarrow A}$ and Multi$_{G \rightarrow M}$ with images containing person bounding boxes of varying sizes. There is a strong positive correlation between bias amplification and the person bounding box size with a Pearson's $r$ of 0.89 and 0.92 for BiasAmp$_{G \rightarrow A}$ and Multi$_{G \rightarrow M}$, respectively (see Fig. 3). This corroborates findings from Hall *et al.* [7] that suggest harder to recognize groups can result in lower bias amplification.

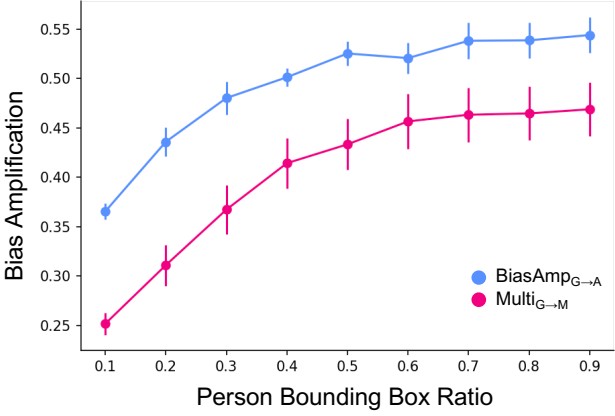

Figure 3: $G \rightarrow A$ bias amplification (calculated using absolute differences) as a function of the person bounding box ratio for COCO. Bounding box ratio is calculated using the area annotation divided by the image dimensions. We plot the average over five runs with random seeds of the model. Error bars represent the standard deviation over runs.

# F  Mitigation methods

We provide a description of the five mitigation strategies used in Sec. 5. In total, we have 18,177, 4,545, and 10,795 images for the train, validation, and test splits with 30.9% of the instances being male. For imSitu, we have 10,240, 6,175, and 24,698 images in each split with 40.7% of each instance being male.

## F.1  Method details

**Oversampling.**  We use a greedy oversampling process to balance group membership for each single attribute. The sampling process terminates when the bias score $b(a, g) \in [|\mathcal{G}|^{-1} \pm \epsilon]$ for all $(a, g)$, where $\epsilon = 0.025$. This results in 45,657 and 40,470 training images for COCO and imSitu respectively. We train a ResNet-50 for 50 epochs using an Adam optimizer with batch size 32, learning rate of $10^{-5}$, and weight decay of $10^{-6}$.

**RBA.**  Reducing Bias Amplification [36] is employed after training. Here, the method uses corpus level constraints so that the predictions match a specified distribution. To ensure that the algorithm converges, the method uses Lagrangian relaxation. We use RBA to optimize the prediction distribution so it matches that of the training set.

**Adversarial Debiasing.**  This method removes group information from the intermediate representation so that the model is not influenced by the group when making predictions. The goal during training is thus to improve the classifier's ability to predict attributes while making it difficult for the adversary to predict group membership.

We remove gendered information from the final convolutional layer of a ResNet-50 (i.e., **adv @ conv5** from Wang *et al.* [30]). To train our adversarial debiasing method, we follow the procedure from Wang *et al.* [30]. We train with the adversarial loss using a learning rate of $5 \times 10^{-6}$ for 60 epochs. Also, we balance the number of female and male images sampled per batch.

**Domain Independent.**  As opposed to adversarial debiasing which promotes *fairness through blindness*, Wang *et al.* [31] claim that domain independent training promotes *fairness through awareness*. Concretely, domain independent training attempts to learn to differentiate between the same attribute for different groups. For example, in the case of gender bias amplification on COCO, the classifier will attempt to learn the boundary between (`computer`, `F`) and (`computer`, `M`). We train the ResNet-50 for 50 epochs using an Adam optimizer with batch size 32, learning rate of $10^{-5}$, and weight decay of $10^{-6}$.

At inference time, there are $2n$ predictions (i.e., $n$ predictions for each group). To select which predictions we use, we take the predictions associated with the ground-truth group membership for the instance. We select this inference method as it gives the highest mAP on the validation set.

**Data Repair.**  Agarwal *et al.* [1] attempt to repair the existing dataset using a fair selection process. Concretely, this method curates a dataset via subsampling until co-occurring attributes are well represented. In the supervised setting, the method uses a greedy approach that aims to minimize $c_v$, which is equal to standard deviation divided by the mean number of images per attribute. We use five random seeds, resulting in a mean coefficient variation of 1.78 and 5.83 for COCO and imSitu. The subsampled train and validation set size 6,000 and 1,000 respectively for both datasets. Here, the models are trained on a smaller number of instances since the method involves subsampling the dataset.

## F.2  Additional results

We provide the results of using different bias mitigation methods on COCO in addition to the results we have already shown on imSitu in Tbl. 6. Looking across datasets, no mitigation method clearly outperforms another. For example, DOMIND outperforms ORIGINAL for COCO on all metrics except for $BiasAmp_{G \to A}$ and $Multi_{G \to M}$; however, the method fares worse on imSitu. This is likely because DOMIND attempts to distinguish between group membership within attributes (e.g., `woman` with `computer` versus `man` with `computer`). Since imSitu attributes, such as indoor or outdoor location, are broadly defined and have diverse appearances compared to objects in COCO, it may be more difficult to learn these boundaries. A potential avenue for inquiry is developing training methods that work well for more general attributes like those found in imSitu.

| (a) COCO | mAP | BiasAmp$_{MALS}$ | BiasAmp$_{A \to G}$ | BiasAmp$_{G \to A}$ | Multi$_{MALS}$ | Multi$_{M \to G}$ | Multi$_{G \to M}$ |
|---|---|---|---|---|---|---|---|
| ORIGINAL | $53.4 \pm 0.2$ | $-0.6 \pm 0.3$ | $2.2 \pm 0.4$ | $-0.0 \pm 0.0$ | $15.9 \pm 0.7$ | $0.4 \pm 0.0$ | $\mathbf{0.2 \pm 0.0}$ |
| OVERSAMPLING | $51.5 \pm 0.1$ | $1.1 \pm 0.1$ | $-3.4 \pm 0.2$ | $-0.2 \pm 0.0$ | $14.4 \pm 0.2$ | $0.3 \pm 0.0$ | $0.3 \pm 0.0$ |
| RBA | $50.7 \pm 1.1$ | $3.8 \pm 1.7$ | $-6.3 \pm 3.5$ | $0.1 \pm 01$ | $15.2 \pm 0.7$ | $7.4 \pm 3.3$ | $0.5 \pm 0.2$ |
| ADV | $\mathbf{59.0 \pm 0.1}$ | $-0.7 \pm 0.9$ | $7.0 \pm 0.6$ | $0.1 \pm 0.0$ | $16.3 \pm 0.4$ | $1.0 \pm 0.2$ | $0.3 \pm 0.0$ |
| DOMIND | $56.1 \pm 0.3$ | $0.4 \pm 0.6$ | $\mathbf{0.0 \pm 0.0}$ | $0.3 \pm 0.0$ | $\mathbf{12.7 \pm 0.5}$ | $\mathbf{0.0 \pm 0.0}$ | $0.3 \pm 0.0$ |
| DATA REPAIR | $48.5 \pm 0.1$ | $\mathbf{0.3 \pm 0.1}$ | $1.9 \pm 0.3$ | $-0.0 \pm 0.0$ | $14.0 \pm 0.3$ | $0.5 \pm 0.0$ | $0.3 \pm 0.0$ |

| (b) imSitu | mAP | BiasAmp$_{MALS}$ | BiasAmp$_{A \to G}$ | BiasAmp$_{G \to A}$ | Multi$_{MALS}$ | Multi$_{M \to G}$ | Multi$_{G \to M}$ |
|---|---|---|---|---|---|---|---|
| ORIGINAL | $67.1 \pm 0.1$ | $2.5 \pm 0.1$ | $-0.3 \pm 0.1$ | $0.0 \pm 0.0$ | $\mathbf{15.1 \pm 0.1}$ | $3.3 \pm 0.1$ | $\mathbf{0.1 \pm 0.0}$ |
| OVERSAMPLING | $66.3 \pm 0.1$ | $-4.5 \pm 0.2$ | $-2.4 \pm 0.1$ | $-0.0 \pm 0.0$ | $15.7 \pm 0.4$ | $0.2 \pm 0.1$ | $0.1 \pm 0.0$ |
| RBA | $54.7 \pm 0.5$ | $\mathbf{-1.4 \pm 0.3}$ | $-6.2 \pm 0.3$ | $-0.1 \pm 0.0$ | $17.6 \pm 0.3$ | $12.4 \pm 0.9$ | $1.1 \pm 0.1$ |
| ADV | $58.1 \pm 0.1$ | $4.1 \pm 0.3$ | $0.6 \pm 0.4$ | $-0.0 \pm 0.0$ | $19.1 \pm 0.5$ | $5.4 \pm 0.8$ | $0.3 \pm 0.1$ |
| DOMIND | $\mathbf{69.6 \pm 0.1}$ | $10.2 \pm 0.9$ | $\mathbf{0.0 \pm 0.0}$ | $0.1 \pm 0.0$ | $20.3 \pm 0.6$ | $\mathbf{0.0 \pm 0.0}$ | $0.5 \pm 0.2$ |
| DATA REPAIR | $62.3 \pm 0.1$ | $-1.8 \pm 0.1$ | $-0.1 \pm 0.1$ | $-0.0 \pm 0.0$ | $21.4 \pm 0.3$ | $14.0 \pm 0.1$ | $1.6 \pm 0.0$ |

Table 6: Comparison of 5 mitigation methods—OVERSAMPLING, RBA, ADV [30], DOMIND, and DATA REPAIR—against the baseline ORIGINAL on COCO and imSitu. We report the $95\%$ confidence interval over 5 models trained using random seeds. The bold values indicate the best performing method: distance to 0 for single-attribute and smallest value for multi-attribute metrics.

| Rank | COCO Single | COCO Multi |
|---|---|---|
| 1 | {motorcycle} | {wine glass,banana,dining table} |
| 2 | {cat} | {car,motorcycle,truck,bench} |
| 3 | {kite} | {car,bus,traffic light,bench} |
| 4 | {couch} | {suitcase,couch,bed} |
| 5 | {microwave} | {spoon,bowl,pizza,microwave} |

| Rank | imSitu Single | imSitu Multi |
|---|---|---|
| 1 | {sowing} | {indoors, wrapping} |
| 2 | {cheering} | {indoors, putting} |
| 3 | {rehabilitating} | {indoors, twirling} |
| 4 | {caressing} | {indoors, manicuring} |
| 5 | {peeing} | {indoors, paying} |

Table 7: Top five groups of attributes with the largest contribution to bias amplification when training and evaluating on a perfectly "balanced" dataset for COCO and imSitu. The ranking is calculated by averaging over the deltas for five runs with random seed assignments. Bolded values indicate the bias is amplifying for male gender expression.

### F.3 Performance on multi-attribute bias amplification

We observe existing bias mitigation methods inadvertently increase multi-attribute bias amplification. For OVERSAMPLING, RBA, and DOMIND, this is expected given that the methods explicitly attempt to address biases with respect to single attributes through balancing, corpus-level constraints, or learned boundaries. If single attributes are balanced, we may expect the model to rely on remaining spurious correlations, such as biases w.r.t multiple attributes. For DATA REPAIR, this method does take into account multiple attribute co-occurrences. However, since DATA REPAIR relies on subsampling, we have to balance between perfectly balance all co-occurrences and the remaining dataset size. Thus, even though some spurious correlations are addressed, it is likely not all multi-attribute co-occurrences are balanced, which the classifier can again exploit.

## G Qualitative results

We consider the top attributes or groups of attributes that contribute to bias amplification. Concretely, we examine the difference in bias scores between predictions and the training set when calculating BiasAmp$_{MALS}$ and Multi$_{MALS}$. The results on the "balanced" datasets from Sec. 4 and mitigation methods from Sec. 5 are found in Tbls. 7 and 8 respectively.

**"Balanced" models.** We find that multi-attributes surfaces a different set of attributes than when only considering for single attributes. In Tbl. 7, there is some overlap between single attributes in the multi-attribute groups (e.g., motorcycle, couch, microwave). However, for imSitu, the multi and single attributes are disjoint.

| Rank | ORIGINAL | OVERSAMPLE | RBA | ADV | DOMIND | DATA REPAIR |
|------|----------|------------|-----|-----|--------|-------------|
| 1 | {umbrella, book} | {horse, phone} | {banana, couch} | {car, umbrella, tie} | {bed, laptop, phone} | {umbrella, ball} |
| 2 | {handbag, bed} | {frisbee, handbag} | {cat, umbrella} | {umbrella, bottle} | {umbrella, book} | {umbrella, TV} |
| 3 | {bike, kite} | {umbrella, TV} | {horse, phone} | {plant, phone} | {car, umbrella, tie} | {umbrella, frisbee} |
| 4 | {bed, book, teddybear} | {tie, suitcase} | {car, umbrella, handbag} | {bus, bench} | {bike, backpack, umbrella} | {umbrella, book} |
| 5 | {fork, cake} | {bike, frisbee} | {handbag, surfboard} | {pizza, couch} | {bus, bench} | {couch, toothbrush} |

| Rank | ORIGINAL | OVERSAMPLE | RBA | ADV | DOMIND | DATA REPAIR |
|------|----------|------------|-----|-----|--------|-------------|
| 1 | {indoors, manicuring} | {indoors, twirling} | {indoors, lathering} | {indoors, manicuring} | {indoors, disciplining} | {indoors, twirling} |
| 2 | {stroking} | {indoors, embracing} | {indoors, manicuring} | {indoors, eating} | {indoors, welding} | {indoors, baptizing} |
| 3 | {indoors, slouching} | {embracing} | {indoors, embracing} | {indoors, spying} | {helping} | {indoors, signing} |
| 4 | {ducking} | {indoors, practicing} | {embracing} | {spying} | {hurling} | {indoors, crushing} |
| 5 | {indoors, sprinting} | {indoors, ducking} | {training} | {training} | {indoors, sleeping} | {indoors, sprinting} |

Table 8: Top five groups of attributes with the largest contribution to bias amplification for bias mitigation methods. Results are reported on COCO (top) and imSitu (bottom). The ranking is calculated by averaging over the deltas for five runs with random seed assignments. Bolded values indicate the bias is amplifying for male gender expression.

**Bias mitigation methods.** Next, we look at the groups surfaced after applying mitigation techniques (Tbl. 8). First, we observe in some cases single attributes can contribute more bias in comparison to their multi-attribute counterparts. For DOMIND, `helping` and `hurling` are in the top five groups of attributes whereas {`helping, indoors`} and {`hurling, indoors`} are not. Second, we note certain groups of attributes occur across many mitigation methods. For example, in COCO, {`umbrella, book`} occurs in DOMIND and DATA REPAIR as well as ORIGINAL. Similarly, in imSitu, {`indoors, manicuring`} also occurs in RBA and ADV as well as ORIGINAL. This indicates there are certain groups of attributes which may be more difficult to debias than others. Learning to address the bias amplification arising from these more difficult groups can be a fruitful direction for future work in this space.

# H  Limitations

We address limitations of our proposed metric.

**Reliance on annotations.** Our proposed metric measures bias amplification w.r.t annotated attributes. This is the same for existing co-occurrence metrics [36, 29]. While we capture multi-attribute bias amplification, we cannot account for amplification occurring with attributes that are not annotated. In addition, for group membership, we rely on third-party annotations, which can be problematic for protected groups. This is a larger problem many fairness-related works in computer vision face [35, 3, 33] due to the lack of self-reported demographic information.

**Entanglement between metrics.** Taking the absolute value of differences conflates two values of the same magnitude but different signages. Both values are equal contributors to bias amplification. Some may argue that a bias amplification score of $-c$ is more desirable than $+c$ since "bias" has decreased. The preference between a positive or negative amplification is a value-laden decision that ultimately depends on the attribute and group. To illustrate, revisiting the example from Sec. 3, negative bias amplification for {`typing, F`} can contribute to erasure. Conversely, {`baking, F`} has a positive bias score of 0.6 in imSitu's train set. Positive bias amplification could mean that harmful social stereotypes are being reproduced. In both cases, bias amplification is undesirable, regardless of direction.

Rather than making an overarching prescription, we propose using a multiplicity of fairness metrics. For example, reporting both raw and absolute differences permits a finer-grained analysis of where amplification is arising.

**Equal weighting of attribute groups.** Similar to existing metrics, we weigh all attribute combinations equally. It may be preferable to weight socially salient attribute combinations more heavily; however, the choice of weights is again context-dependent. For example, if a classifier trained on COCO is deployed in an athletics setting, amplification wrt sports attributes such as `frisbee` or `tennis racket` may be more important. Alternatively, if the classifier is deployed in a classroom setting, these attributes may not be as salient. We suggest metric users consider the deployment context of their system and adjust weights accordingly.

**Runtime analysis.** Naively implemented, the runtime for calculating the multi-attribute metric could be exponential as we could iterate through all possible combinations of attributes $\mathcal{A}$ of size $k$ for $k \in \{1, \ldots |\mathcal{A}|\}$. This is especially costly given that $|\mathcal{A}|$ can be large for many visual datasets. However, we only consider the groups of multiple attributes that exist in the dataset; many groups of multiple attributes do not occur in either the training or the test set. Rather, our implementation can run in $\mathcal{O}(n)$ time where $n$ is equal to the number of instances in either the training or test set.

