# OpenReview forum: "Men Also Do Laundry: Multi-Attribute Bias Amplification"
_NeurIPS.cc/2022/Workshop/TSRML — TSRML2022_

### Official Review · Reviewer_XhKJ · 2022-10-20
**Interesting Concept, not well enough explored**

**Overall Rating:** 5

**Summary:**

This paper provides:
1) A description of a new type of bias amplification.
2) Mathematical and real world examples showing when it can occur.
3) A formal definition for the multi-attribute bias amplification metric.
4) A series of experiments comparing various design choices, and measuring the amplification in practive.

**Strengths:**

The paper provides and interesting extension to the bias amplification concept. They show a real world example of where it occurs
(ie unloading laundry indoors vs unloading a car/etc). They define mathematically their new metric, and compare it to the existing bias amplification metric.

**Weaknesses:**

The authors do not thoroughly explore how group size effects the measurement result.  They only show examples for k>=1 and k>=2. To my knowledge there is no upper bound on the number of the group size. If I misinterpret this from the paper, please clarify. I would expect that larger group sizes would have larger scores for bias amplification, as the number of examples would decrease as the group size increase, as the problem becomes increasingly constrained. This can create a large amount of noise in the metric, especially at large group sizes. The authors do not address how group size will effect number of training/test examples per group, and do not investigate thoroughly for many values of k how the metric changes. Additionally, it would be interesting to only compute bias amplification at a fixed k, either k=2 or k=3. As k increases, the number of real world examples where this bias would be harmful likely decreases.

The title of the paper 'Men Also Do Laundry' does not incorporate the multi-attribute part of their metric.

**Overall Recommendation:**

While multi-attribute bias amplification is a very interesting concept, and the authors have shown interesting scenarios where it can occur, I do not think they have done a thorough enough job into investigating the metric and how the group size can effect the score.

**Review Confidence:**

3: The reviewer is fairly confident that the evaluation is correct

---

### Official Review · Reviewer_KNJ8 · 2022-10-21
**a new metric on multi-attribute debiasing**

**Overall Recommendation:** see above
**Overall Rating:** 6

**Summary:**

This paper proposes a new metric on multi-attribute debiasing

**Strengths:**

1. The idea is novel. An interesting metric is proposed on multi-attribute debiasing, which is a relatively new topic.
2. Detailed analysis is provided.
3. Well-motivated and the writing is good.

**Weaknesses:**

1. Technical contribution is limited. No new debiasing method on multi-Attribute debiasing is proposed.

**Review Confidence:**

3: The reviewer is fairly confident that the evaluation is correct

---

### Official Review · Reviewer_t2eB · 2022-10-22
**Review for "Men also do laundry: multi-attribute bias amplification"**

**Overall Rating:** 7

**Summary:**

Existing studies focused on evaluating the bias amplification of a training set induced by a single attribute. This paper first explores the multi-attribute bias amplification problem via a novel metric, and unveils the limitation of existing single-attribute metrics as well as the bias-mitigation methods in such multi-attribute situations.

**Strengths:**

1. This paper first investigates the multi-attribute bias amplification problem, which is an important but under-explored research question by the previous literature.

2. The experiments on both synthetic and real-world multi-attribute datasets show that the new metric effectively captures the amplified bias by multi-attributes. In addition, its aggregation of the absolute differences allows a better understanding of the metric compared to the single-attribute metrics.

3. The proposed metric reveals the limitations of the existing debiasing methods on multi-attribute bias amplification, leaving the crucial question for future research.


**Weaknesses:**

1. The group membership in this paper is restricted to the gender. Additional experiments on other types of the group memberships, which may also include the non-binary ones, are required to generally understand the effectiveness of the proposed metrics.

2. This paper shows that applying the single-attribute methods results in amplifying the bias in multi-attribute settings, but it does not provide an explanation for why this occurs.

**Overall Recommendation:**

The exploration of the multi-attribute bias amplification is inspiring and reveals the under-explored limitations of existing single-attribute metrics as well as the bias mitigation methods. However, it would be more intriguing if the authors could include additional in-depth study of various types of groups as well as a discussion on existing bias-mitigation methods’ failures. Still, I consider this paper’s contributions to be greater than its weaknesses, thus I’d want to rate it as an accept.

**Review Confidence:**

3: The reviewer is fairly confident that the evaluation is correct

---

### Decision · Program_Chairs · 2022-10-23

Accept